# Chicoric Acid Ameliorated Beta-Amyloid Pathology and Enhanced Expression of Synaptic-Function-Related Markers via L1CAM in Alzheimer’s Disease Models

**DOI:** 10.3390/ijms25063408

**Published:** 2024-03-17

**Authors:** Ruonan Wang, Shijia Kang, Zirui Zhao, Lingling Jin, Xiaolin Cui, Lili Chen, Melitta Schachner, Sheng Li, Yanjie Guo, Jie Zhao

**Affiliations:** 1National-Local Joint Engineering Research Center for Drug-Research and Development (R&D) of Neurodegenerative Diseases, Dalian Medical University, Dalian 116044, China; 18522253613@163.com (R.W.); k15236692297@163.com (S.K.); 13130496510@163.com (Z.Z.); jinlingling820129@163.com (L.J.); sw_cxl@126.com (X.C.); chenlili03070526@hotmail.com (L.C.); lisheng_1996@163.com (S.L.); 2School of Integrated Chinese and Western Medicine, Dalian Medical University, Dalian 116044, China; 3School of Basic Medical Sciences, Dalian Medical University, Dalian 116044, China; 4Keck Center for Collaborative Neuroscience and Department of Cell Biology and Neuroscience, Rutgers, The State University of New Jersey, Piscataway, NJ 08854, USA; schachner@dls.rutgers.edu; 5Department of Biochemistry, School of Basic Medical Science, Dalian Medical University, Dalian 116044, China; 6Department of Microecology, School of Basic Medical Science, Dalian Medical University, Dalian 116044, China

**Keywords:** chicoric acid, amyloid-β plaques, L1CAM, Alzheimer’s disease, synaptic plasticity

## Abstract

Alzheimer’s disease (AD) is the most common progressive neurodegenerative disease. The accumulation of amyloid-beta (Aβ) plaques is a distinctive pathological feature of AD patients. The aims of this study were to evaluate the therapeutic effect of chicoric acid (CA) on AD models and to explore its underlying mechanisms. APPswe/Ind SH-SY5Y cells and 5xFAD mice were treated with CA. Soluble Aβ1–42 and Aβ plaque levels were analyzed by ELISA and immunohistochemistry, respectively. Transcriptome sequencing was used to compare the changes in hippocampal gene expression profiles among the 5xFAD mouse groups. The specific gene expression levels were quantified by qRT-PCR and Western blot analysis. It was found that CA treatment reduced the Aβ1–42 levels in the APPswe/Ind cells and 5xFAD mice. It also reduced the Aβ plaque levels as well as the APP and BACE1 levels. Transcriptome analysis showed that CA affected the synaptic-plasticity-related genes in the 5xFAD mice. The levels of L1CAM, PSD-95 and synaptophysin were increased in the APPswe/Ind SH-SY5Y cells and 5xFAD mice treated with CA, which could be inhibited by administering siRNA-L1CAM to the CA-treated APPswe/Ind SH-SY5Y cells. In summary, CA reduced Aβ levels and increased the expression levels of synaptic-function-related markers via L1CAM in AD models.

## 1. Introduction

Alzheimer’s disease (AD) is a neurodegenerative disease that is clinically characterized by memory and cognitive dysfunctions, accompanied by the deterioration of a person’s speech, visuospatial skills, emotionality and overall personality [1]. According to the World Health Organization (WHO), by 2030, the number of people with AD will increase to 82 million, and dementia will become a global crisis [2]. The pathological features of AD include the accumulation of Aβ plaques and the hyperphosphorylation of tau protein [3]. Amyloid-β is formed by the cleavage of amyloid precursor proteins (APPs) via β-secretase (BACE1) and γ-secretase. Abnormal APP processing and the release of neurotoxic Aβ fragments cause Aβ to aggregate into oligomers, which then polymerize to form fibrils that produce amyloid plaques [4]. Soluble Aβ oligomers and amyloid plaques interact with glia cells and neurons to induce impaired cellular responses that ultimately lead to neuronal cell death [5]. Therefore, the development of potential therapeutic agents targeting Aβ pathology may reveal beneficial effects on AD.

Traditional Chinese herbs contain molecules that are likely to be effective ingredients for the treatment of AD [6], among which chicoric acid (CA) is a promising candidate. CA belongs to the groups of hydroxycinnamic acids and phenylpropanoids commonly found in plants such as cichorium intybus, echinacea purpurea and other members of the asteraceae family [7,8], and it is widely used as a nutrition supplement [9]. It has exhibited extensive pharmacological effects, such as regulating glycolipid metabolism and reducing inflammation, oxidation and aging [10]. It was shown that CA prevented dopaminergic neuropathy, motor deficit and glial activation in N-methyl-4-phenyl-1,2,3,6-tetrahydropyridine (MPTP)-induced Parkinson’s disease (PD) in mice [11]. CA has also displayed neuroprotective effects in a mouse model of PD via modulation of the gut microbiota and TLR-4 signaling pathway [12]. CA alleviated NLRP3-mediated pyroptosis in an acute-lung-injury model through ROS-induced mitochondrial damage [13]. Moreover, CA alleviated memory impairment and amyloid formation by inhibiting the NF-κB transcriptional pathway in a lipopolysaccharide (LPS)-induced systemic inflammation mouse model [9]. These findings suggest that CA may be a viable therapeutic intervention for neurodegenerative diseases such as Alzheimer’s disease.

Synapses are critical for brain neuron communication. It was shown that pathological Aβ accumulation in synapses mediated synaptic loss and dysfunction [14]. L1CAM, hereon referred to as L1, is a member of the immunoglobulin superfamily of adhesion molecules and is highly expressed in the nervous system and is involved in synapse formation, synaptic activity, plasticity and synaptic vesicle circulation at different developmental stages [15]. Adenoviruses encoding full-length L1 were injected into the hippocampus and cortex of AD APP/PS1 model mice. After four months, a reduced Aβ1–42 level, decreased Aβ1–42/40 ratio and fewer β-amyloid plaques were observed in the mouse brains [16]. Given the important roles of L1 in the nervous system, molecules in traditional Chinese medicines that activate the L1 effect are expected to have therapeutic effects on AD. Therefore, in this study, we investigated the effects of CA via L1 on Aβ pathology and synaptic markers in APPswe/Ind transgenic SH-SY5Y cells and 5xFAD mice. The study provides evidence for the future application of chicoric acid in the treatment of AD.

## 2. Results

### 2.1. CA Reduced Aβ1–42, APP and BACE1 Expression Levels in APPswe/Ind Transgenic SH-SY5Y Cells

To investigate the effect of CA on the level of Aβ1–42 in APPswe/Ind transgenic SH-SY5Y cells, the concentration of Aβ1–42 was measured. The results showed that the Aβ1–42 level was higher in the APPswe/Ind transgenic SH-SY5Y cells than it was in the non-APPswe/Ind transfected SH-SY5Y cells. CA treatment reduced the Aβ1–42 level in the APPswe/Ind transgenic SH-SY5Y cells (Figure 1A). Furthermore, we compared the non-APPswe/Ind cells with the transfected SH-SY5Y cells. The mRNA levels of APP and BACE1 were higher in the APPswe/Ind transgenic SH-SY5Y cells, and CA treatment decreased the APP and BACE1 levels in the SH-SY5Y-APPswe/Ind cells (Figure 1B–F).

### 2.2. CA Decreased the Amyloid-β Plaque Load and Levels of APP and BACE1 in 5xFAD Mice

To investigate the role of CA in vivo, we analyzed the effect of CA on Aβ plaque levels in the 5xFAD mice. The results showed that there were many Aβ plaques deposits in the hippocampi of these mice, and CA treatment decreased the number of Aβ plaques and the Aβ1–42 level (Figure 2A,B). The APP and BACE1 levels were higher in the 5xFAD mice than in the WT group, while CA treatment decreased their expression levels (Figure 2C–E).

### 2.3. Transcriptomic and Bioinformatic Analyses of the Hippocampi in 5xFAD Mice

Transcriptomic analysis was used to analyze the effect of CA treatment on the hippocampus gene changes in 5xFAD mice. As shown in Figure 3A–C, 19 synaptic-related genes were identified that differed in expression among the WT, 5xFAD and CA-treated 5xFAD mice. Further analysis using the GO database showed that these differently expressed genes were involved in synaptic functions such as neuron-to-neuron synapse, postsynapse and overall synapse organization. These gene-expression profiles of CA-treated mice indicated that CA improved synaptic function in 5xFAD mice.

### 2.4. CA Increased Synaptic Density in APPswe/Ind Transgenic SH-SY5Y Cells

Synaptic proteins are key regulators in the dynamics of presynaptic terminal synaptic vesicles and can regulate synaptic transmission. Synaptophysin is a presynaptic marker. Postsynaptic density protein 95 (PSD-95), is a postsynaptic marker and the main synaptic scaffold molecule regulating synaptic maturation. Synaptophysin and PSD-95 are important for synaptic function. We explored the effects of CA on these molecules, and the results in Figure 4A–C showed that the levels of synaptophysin and PSD-95 were decreased in the APPswe/Ind transgenic SH-SY5Y cells when compared with those in the SH-SY5Y control cells. CA treatment reversed these changes.

### 2.5. CA Enhanced Hippocampal Synaptic Density in 5xFAD Mice

CA increased the synaptic density in the 5xFAD mice. As shown in Figure 5A–C, compared with the WT control group, the levels of synaptophysin and PSD-95 were decreased in the 5xFAD mice, and CA treatment increased levels of these proteins.

### 2.6. CA Treatment Increased L1 Levels in APPswe/Ind Transgenic SH-SY5Y Cells and 5xFAD Mice

Previous studies have reported that the neural cell adhesion molecule L1 binds to Aβ and treated this pathology in adult mouse brain tissue [16]. To study the effect of CA on L1 expression, we measured L1 levels in APPswe/Ind transgenic SH-SY5Y cells and in 5xFAD mice after CA treatment. The results in Figure 6A–D showed that the expression level of L1 was lower in the APPswe/Ind transgenic SH-SY5Y cells and the 5xFAD mice when compared with the control. CA treatment increased the L1 levels both in vitro and in vivo.

### 2.7. siL1-RNA Affects CA-Regulated Levels of AD- and Synapse-Related Proteins in APPswe/Ind Transgenic SH-SY5Y Cells and 5xFAD Mice

We tested whether siL1 affects the outcome of CA treatment on the levels of Aβ and synaptic density protein, and confirmed that siL1-RNA reduced L1 levels (Figure 7A,B). As shown in Figure 7C–F, Aβ1–42 levels were increased in the non-treated APPswe/Ind transgenic cells compared with those of the SH-SY5Y cells. In the APPswe/Ind transgenic SH-SY5Y cells, the expression levels of APP and BACE1 were increased, whereas the levels of PSD-95 and synaptophysin were decreased. CA treatment reversed these effects in the APPswe/Ind transgenic cells. However, pre-treatment with siL1 RNA affected the levels of these molecules induced by CA in the APPswe/Ind transgenic SH-SY5Y cells. Thus, L1 was involved in mediating the effects of CA on cells expressing APP and BACE1 or PSD-95 and synaptophysin.

## 3. Discussion

The threat of AD to human society is rapidly increasing as the number of senior people rises steadily around the world. The current drugs used to treat AD consist mainly of cholinesterase inhibitors and glutamate receptor antagonists [17]. However, it is noteworthy that these drugs have limited efficacy and chronic use of them leads to serious adverse effects [18,19]. To date, there have been no curative treatments available that can effectively halt the occurrence of or slow the progression of AD. Potential treatments for AD are still urgently needed by the medical society.

Traditional Chinese medicine exerts beneficial effects on many chronic diseases, possibly due to the fact that the active substances in traditional Chinese medicine have the characteristics of multiple therapeutic effects. Herein, we focus on the effect of CA, which has exhibited various bioactivities [9], and played beneficial roles on Aβ pathology and synaptic plasticity in the present study. Aβ plaques are a distincive pathological feature in AD patients and were once considered the most promising target for AD treatment. To date, several Aβ-plaque-reducing agents such as lecanemab and aducanumab have been approved by the FDA for the treatment of AD [20]. However, it should be noted that there is controversy regarding the role of Aβ plaques in the progression of AD. The participation of amyloids in neuronal death has not yet been demonstrated in vivo. A huge body of evidence has shown that Aβ overproduction is associated with the breakdown of energy metabolism in the brains of both humans and animals [21,22,23]. The role of Aβ and amyloids in AD requires further investigation.

Previous studies have shown that CA has inhibited the accumulation of Aβ and BACE1 expression in LPS-induced systemic inflammation in mouse brains and improved mitochondrial function and regulated energy metabolism, thus protecting neurons from inflammation [24]. In the present study, we examined the effect of CA on Aβ pathology, and synaptic formation. We found that CA reduced the expression of APP and BACE1 and decreased Aβ1–42 levels in APPswe/Ind SH-SY5Y cells and in 5xFAD mice. CA also reduced the aggregation of Aβ plaques in the hippocampus of 5xFAD mice. Synaptic dysfunction is one of the earliest pathological changes in the brains of AD patients. Excess deposition of Aβ disrupts synaptic plasticity and mediates synaptic toxicity through different mechanisms [25]. In this study, transcriptomic analysis was performed to screen the differential expression of genes related to synaptic function among the wild-type and 5xFAD mice or CA-treated 5xFAD mice. Nineteen genes were identified to be differentially expressed between the groups. These genes are involved in synaptic functions including neuron-to-neuron synapse and postsynapse and synapse organization. The analyses of gene interactions profiles showed that proteins with high interactions were enriched in the synaptic adhesion-like-molecule pathway and the glutamatergic-synapse pathway, the unblocking of NMDA receptors and the glutamate-binding and activation pathway (Appendix A).

PSD-95 acts as a scaffold protein in postsynaptic density, organizing signaling molecules by contributing to synaptic transmission and synaptic plasticity. Synapses contain large amounts of endogenous PSD-95, which could protect against Aβ toxicity [26]. However, PSD-95 was reduced in the brain tissue of patients with AD [27]. Synaptophysin, as a presynaptic biomarker, exists on the synaptic vesicular membrane and regulates synapse formation. A study that analyzed and compared the protein levels in frontal and parietal cortices of a high number of patients with AD and healthy controls by immunoblot analysis showed that all brain specimens from patients with AD had lost synaptophysin [28]. Increasing the expression of postsynaptic and presynaptic proteins may improve synaptic function. In this study, our data show that expression of PSD-95 and synaptophysin was reduced in 5xFAD mice and in APPswe/Ind cells compared with the controls, while CA treatment increased the expression of PSD-95 and synaptophysin. This finding indicates that CA treatment may improve synaptic plasticity in an AD mouse model.

Previous studies have shown that Aβ peptides bind to L1 in vitro via its second Fn3 domain, and this interaction reduces the formation of high-molecular-weight forms of Aβ. An in vivo study has shown that the hippocampal L1 levels were reduced in aged APPswe mice, leading to Aβ deposition. The L1 fragment (L1–70), generated by serine proteases, hydrolyzed L1, activated microglia and induced cytokine expression and the clearance of Aβ plaques [29]. These findings suggest a protective role for L1 in AD via the inhibition of Aβ aggregation and the promotion of Aβ clearance. Consistent with this, our data show that L1 expression was significantly reduced in 5xFAD mice and APPswe/Ind cells. CA treatment promoted the expression of L1 in the APPswe/Ind SH-SY5Y cells and 5xFAD mice. L1 regulates synaptic activity by recruiting function-relevant molecules to the synapses, regulating synaptic vesicular exocytosis and endocytosis and inducing intracellular signaling cascades [30]. By using siRNA against the L1 gene, the study further explored the role of CA in inhibiting Aβ aggregation and improving synaptic plasticity and its dependence on up-regulation of the L1 gene. The inhibition of L1 reversed these effects of CA in the APPswe/Ind SH-SY5Y cells. These data indicate that CA alleviated Aβ pathology and enhanced the expression of synaptic-function-related markers via L1.

Previous studies have shown that the therapeutic effect of CA on neurodegenerative diseases may be related to several mechanisms: (1) anti-inflammatory and anti-oxidative effects—CA prevented LPS-induced systemic-inflammation-induced memory impairment via inhibition of NF-κB [9] and (2) neuronal protective effects—CA improved neuron mitochondrial function and energy metabolism [31]. The present study showed that CA also alleviated Aβ pathology and improved the expression of synaptic-function-related markers via L1. It is noted that some active substances in traditional Chinese herbs, such as chlorogenic acid and ursolic acid, have similar roles to CA in the treatment of neurodegenerative disease [32,33]. Chlorogenic acid improved mitochondrial dysfunction and protected against neuronal cell death in a PD mouse model [34]. Ursolic acid exerted strong antioxidant properties and improved behavior in these mice. Ursolic acid attenuated oxidative stress in nigrostriatal tissue and improved behavior in a MPTP-induced PD mouse model [35]. It is implied that active substances of traditional Chinese herbs exhibiting anti-inflammatory, anti-oxidant and other neuroprotective properties may be potential therapeutic candidates for the treatment of neurodegenerative diseases. They usually act in multiple pathways, and these require extensive investigation.

## 4. Materials and Methods

### 4.1. Chemicals and Reagents

Chicoric acid was obtained from Shanghai Yuanye Bio-Technology (B20647, Shanghai, China). The BCA Protein Determination Kit and protein lysis buffer were from Beyotime (P0009, Shanghai, China). The anti-L1CAM (1:1000, ab270455) and anti-synaptophysin antibodies (1:1000, ab16659) were from Abcam (Cambridge, UK). The anti-PSD-95 (1:1000, 3450S) was from Cell Signaling Technology (Boston, MA, USA). The anti-APP antibodies (1:1000, 60342-1-Ig) was from proteintech (Wuhan, China). The BACE1 antibody (1:1000, A5095) was from Selleck (Houston, TX, USA). Goat anti-mouse (1:1000, 31430) or goat anti-rabbit (1:1000, 31460) IgG-HRP antibodies were from Thermo Fisher Scientific (Waltham, MA, USA). Lipofectamine 3000 (L3000150) was from Thermo Fisher Scientific (Waltham, MA, USA). si-L1 (5′-3′: GCAAGAUCUUGCACAUCAATT; 5′-3′: UUGAUGUGCAAGAUCUUGCTT) was from Gene Pharma Life Technologies (Shanghai, China).

### 4.2. Cell Culture and Treatment

The SH-SY5Y cells were obtained from Zhejiang University (CL-0595, Procell Life Science & Technology, Wuhan, China). The human APP Swedish/Indiana gene (APPswe/Ind) was constructed in the vector (Appendix A), and an empty vector was used as a control. To obtain stable cell lines, single-cell clones were generated by selection with 5 μg/mL blasticidin S (S7419, Selleck, Houston, TX, USA). The cells were cultured in Dulbecco’s modified Eagle’s medium (11965118, Gibco, Waltham, MA, USA) containing 10% fetal bovine serum (A5669701, Gibco, Waltham, MA, USA) and 1% penicillin–streptomycin solution (15140122, Thermo, Waltham, MA, USA) at 37 °C in a humidified atmosphere and 5% CO_2_. The APPswe/Ind SH-SY5Y cells could overproduce Aβ; therefore, the factors influencing soluble Aβ1–42 levels could be investigated.

The CCK-8 cell viability experiment was used to analyze the effects of different doses of CA on SH-SY5Y cell viability. As shown in Appendix A, CA had no effect on cell viability when it was less than 400 μM. In the experiments on siRNA L1 inhibition, the SH-SY5Y cells were divided into four groups: (1) SH-SY5Y (serum-free DMEM), (2) SH-SY5Y-APPswe/Ind, (3) SH-SY5Y-APPswe/Ind + CA (80 μM) and (4) SH-SY5Y + CA (80 μM) + si-L1. siRNAL1 was co-transfected into the SH-SY5Y-APPswe/Ind cells using lipofectamine 3000 [14].

### 4.3. Mice and Treatments

All the animal care and experimental procedures were in accordance with the principles and guidelines of the National Institutes of Health Guide for the Care and Use of Laboratory Animals and were approved by the Institutional Ethics Committee of Dalian Medical University (IACUC number: AEE23043). This animal study has been reported in compliance with the ARRIVE guidelines.

The mice were housed under standard conditions with food and water available ad libitum. The 5xFAD mice (Tg6799, Jackson Laboratory, Bar Harbor, ME, USA) were mated with the B6/SJL mice (100012, Jackson Laboratory), and the genotypes were determined using RT-PCR analysis. Non-transgenic male mice were used as the wild-type (WT) mice, in parallel with only the male 5xFAD mice. The mice were assigned to four groups (*n* = 10/group): the intragastrically injected WT (phosphate buffer solution (PBS)) group, the 5xFAD (PBS) group, the 5xFAD + CA group (40 mg/kg) group, and the WT + CA (40 mg/kg) group [11]. The CA dose of 40 mg/kg was chosen based on previous studies which showed that CA induces a therapeutic effect, and that there is no toxicity at this dose. Six-week-old mice received CA or PBS once daily for six weeks. The hippocampal tissues were then removed and stored at −80 °C.

### 4.4. Western Blot Analysis

Proteins were extracted from the cells or hippocampus tissues using radio immunoprecipitation assay lysis buffer (P0013B, Beyotime, Shanghai, China) mixed with 1% protease inhibitor (78429, Thermo Fisher Scientific, Waltham, MA, USA) and phosphatase inhibitor (78440, Thermo Fisher Scientific, Waltham, MA, USA). The protein concentration was measured using a BCA Kit (P0009, Beyotime, Shanghai, China). The protein samples were separated using SDS-PAGE, and then transferred to a PolyVinylideneFluoride (PVDF) membrane (IPVH07850, Waltham, MA, USA). Next, the membranes were treated with Superblocking Buffer (37515, Thermo Fisher Scientific, Waltham, MA, USA) for 30 min at 25 °C. The membranes were incubated with primary antibodies at 4 °C overnight. After washing with Tris-buffered saline and Tween 20, the membranes were treated with secondary antibody labeled as HRP (ab6721, Abcam, Cambridge, UK) for 2 h at 25 °C. The membranes were covered with ECL reagents (P0018S, Beyotime, Shanghai, China), and images were taken using an imaging system (Bio-Rad, Hercules, CA, USA). The density of protein was measured and quantitatively analyzed by Image Lab (4.0) software.

### 4.5. Quantitative Real-Time Polymerase Chain RDXeaction (qRT-PCR)

Total RNA samples were isolated from the SH-SY5Y cells by using TRIzol reagent (15596026, Thermo Fisher Scientific, Waltham, MA, USA). After quantification, cDNA was synthesized according to the manufacturer’s instructions (Life Technologies, Carlsbad, CA, USA). cDNA was used for qRT-PCR conducted with an Eppendorf MasterCycler RealPlex4 (Eppendorf, Wesseling-Berzdorf, Germany) using an Ultra SYBR Mixture kit (Q111-02, Vazyme, Nanjing, China). The relative expression of mRNA was normalized to GAPDH and calculated using the 2^−ΔΔCT^ method. Gene primers are shown in Table 1.

### 4.6. Immunohistochemistry

The brains of the mice were perfused transcardially with pre-cooled saline and 4% formaldehyde. Then, the brains were maintained in 4% formaldehyde at 4 °C for 24 h, dehydrated in 30% sucrose and stored at 4 °C. Coronal brain sections of 30 µm thickness were cut with a Leica cryostat microtome (CM1950, Leica, Wetzlar, Germany). The sections were washed with PBS and incubated with 1% H_2_O_2_ for 30 min at room temperature. After being washed with PBS, they were pre-incubated in 0.4% Triton X-100 (85111, Thermo Fisher Scientific, Waltham, MA, USA) and 1% BSA and 4% horse serum in 0.01 M PBS for 1 h at RT. Then, the slices were incubated overnight at 4 °C using primary antibodies against Aβ (1:200, bs-0107R, BISSO, Beijing, China). The slices were washed in PBS, and then treated with HRP-conjugated secondary antibodies (ZK1030, Vector, Shenzhen, China) for 1 h at room temperature. After this, the slices were washed in PBS and incubated with ABC Kits (ZK1030, Vector, Shenzhen, China) for 1 h at RT. The slices were then washed with PBS and incubated with 3-3′ diamionbenzidine (34002, Thermo Fisher Scientific, Waltham, MA, USA). Finally, the slices were scanned using a digital slide scanner KF-PRO-020 (Kfbio, Ningbo, China). Densitometry was analyzed using Image-Pro Plus 6.0 software (Media Cybernetics, Inc., Rockville, MD, USA).

### 4.7. Enzyme-Linked Immunosorbent Assay (ELISA) Analysis

The SH-SY5Y cells and mouse brains were homogenized in lysis buffer (P0013B, Beyotime, Shanghai, China), and centrifuged at 12,000× *g* for 10 min. The soluble Aβ1–42 levels in cells and mouse hippocampi were detected according to the manufacturer’s protocols by using commercial sandwich ELISA kits (E-EL-H0543 and E-EL-M3010, Elabscience, Wuhan, China). The OD values at 450 nm of each group were measured by using EnSpire ELISA (PerkinElmer, Waltham, MA, USA). The concentration of Aβ1–42 was calculated according to the standard curve.

### 4.8. Bioinformatics Analysis

The mice were perfused with PBS, and their hippocampal tissues were dissected on ice. The total mRNA of the hippocampus was removed, and a cDNA library was constructed. Individual samples from the library were sequenced at pair-end 150 bp with the Illumina Hiseq 4000 platform. STAR was employed to map the pair-end reads to a mouse reference genome (https://www.gencodegenes.org/mouse/ (19 September 2023)) and to calculate the read count of each gene. Differential gene expression analysis was performed using DESeq2 of the 5xFAD vs. control groups and 5xFAD + CA vs. 5xFAD groups. DESeq2 provides statistical routines for determining differential levels in digital gene expression using a model based on the negative binomial distribution. The resulting *p*-values were adjusted using the Benjamini and Hochberg approaches for controlling false values. Genes with an adjusted *p*-value ≤ 0.05 found with DESeq2 were assigned as differentially expressed. For each sequenced library, the read counts were adjusted using the edgeR program package prior to differential gene expression analysis through one scaling normalized factor. Differential expression analysis of two conditions was performed using the edgeR R package (3.22.5). The *p* values were adjusted using the Benjamini and Hochberg methods. A corrected *p*-value of 0.05 and a fold change of 2 were set as the threshold for significant differential expression. After this analysis, the up- and down-regulated genes were identified. Gene ontology (GO) enrichment analysis of differentially expressed genes was implemented by the cluster Profiler R package, in which gene length bias was corrected. GO terms with corrected *p* values of less than 0.05 were considered significantly enriched by differentially expressed genes. The GO enrichment analysis of genes was carried out using NovoMagic smart cloud platform (https://magic.novogene.com (29 December 2023)).

### 4.9. Statistical Analysis

All data are presented as mean ± SEM. One-way analysis of variance or two-way analysis of variance was used for data analyses with GraphPad Prism 8.0.1 statistical software. *p* < 0.05 represents a statistically significant difference.

## 5. Conclusions

To the best of our knowledge, this study is the first to describe the effect of chicoric acid in ameliorating Aβ pathology via reduced APP and BACE1 protein expression in 5xFAD mice and APPswe/Ind SH-SY5Y transgenic cells, while improving the expression of synaptic-function-related markers. The therapeutic effect of chicoric acid may be related to its regulation of APP-related secretases and synapse-associated proteins through the up-regulation of L1 expression. This provides a new perspective for chicoric acid to alleviate the pathological process of Alzheimer’s disease via the adhesion molecule L1, and it is a promising drug for the treatment of Alzheimer’s disease.

## Figures and Tables

**Figure 1 ijms-25-03408-f001:**
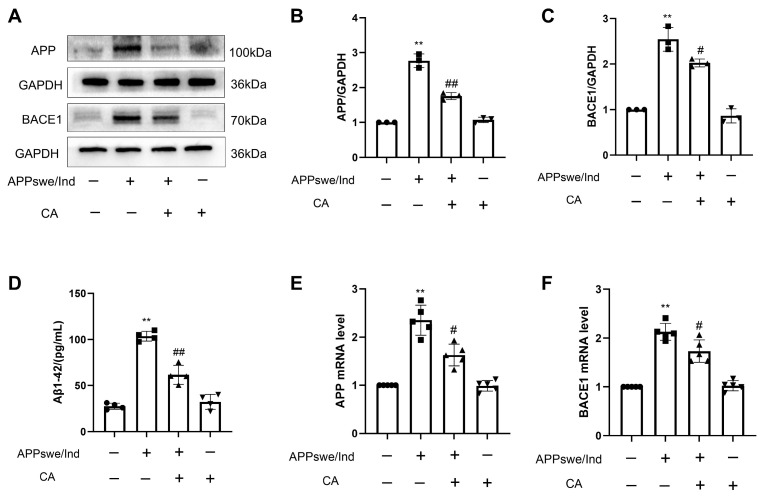
CA reduced Aβ1–42, APP and BACE1 levels in APPswe/Ind transgenic SH-SY5Y cells. (**A**) Level of Aβ1–42. (**B**) mRNA level of APP. (**C**) mRNA level of BACE1. (**D**) Protein bands of APP and BACE1. (**E**) Protein quantification of APP. (**F**) Protein quantification of BACE1. CA: 80 μM; *n* = 3–5. Comparison with the control group, ** *p* < 0.01; comparison with the APPswe/Ind group, ^#^
*p* < 0.05, ^##^ *p* < 0.01.

**Figure 2 ijms-25-03408-f002:**
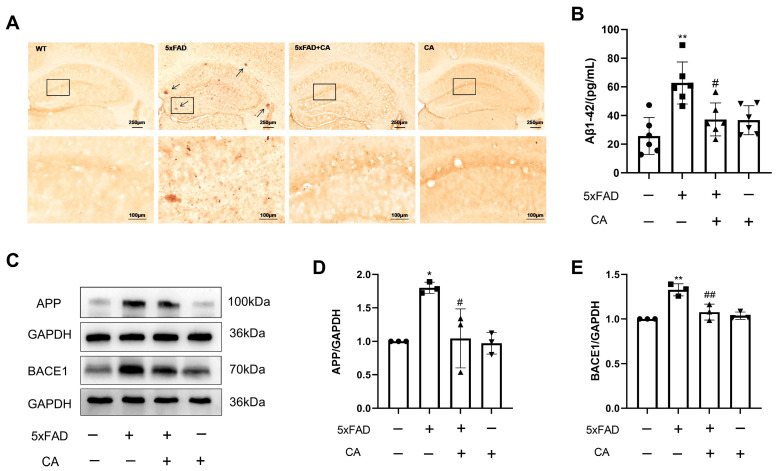
CA reduced levels of amyloid-β plaque deposition and APP and BACE1 expression in 5xFAD mice. (**A**) Amyloid-β plaques in the hippocampi of 5xFAD mice. The squares show the enlarged areas, and the arrows show Aβ plaques. (**B**) Levels of Aβ1–42 in the hippocampi of 5xFAD mice. (**C**) Protein bands of APP and BACE1. (**D**) Protein quantification of APP. (**E**) Protein quantification of BACE1. CA: 40 mg/kg; *n* = 3–5. Comparison with the WT group, * *p* < 0.05, ** *p* < 0.01; comparison with the 5xFAD group, ^#^
*p* < 0.05, ^##^ *p* < 0.01.

**Figure 3 ijms-25-03408-f003:**
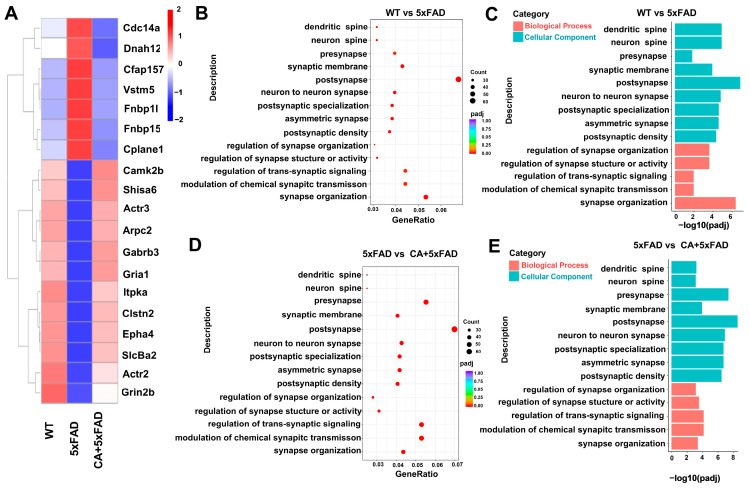
Transcriptomic and bioinformatic analyses of the hippocampus in 5xFAD mice. (**A**) Heat map of different genes. Red and blue indicate high-abundance-level and low-abundance-level genes, respectively. (**B**,**C**) GO enrichment analysis of differently expressed genes of WT and 5xFAD mice. (**D**,**E**) GO enrichment analysis of differently expressed genes in 5xFAD mice and CA-treated 5xFAD mice.

**Figure 4 ijms-25-03408-f004:**
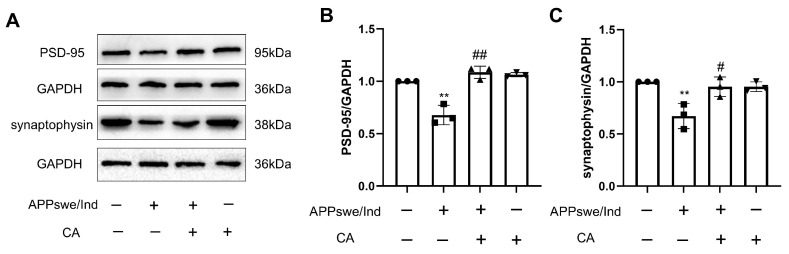
CA increased synaptic density in APPswe/Ind transgenic SH-SY5Y cells. (**A**) Protein bands of PSD-95 and synaptophysin. (**B**) Protein quantification of PSD-95. (**C**) Protein quantification of synaptophysin. CA: 80 μM; *n* = 3. Comparison with the SH-SY5Y control cells, ** *p* < 0.01; comparison with APPswe/Ind transgenic SH-SY5Y cells, ^#^ *p* < 0.05, ^##^
*p* < 0.01.

**Figure 5 ijms-25-03408-f005:**
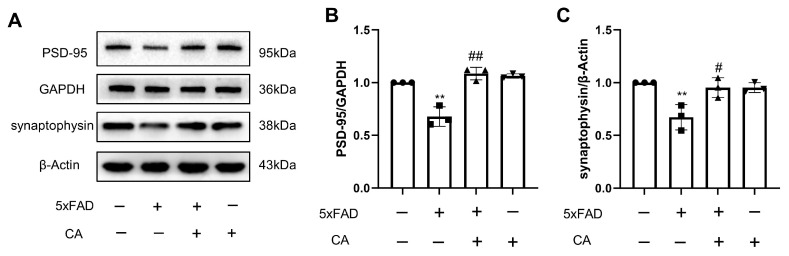
CA increased synaptic density levels in the hippocampus of 5xFAD mice. (**A**) Protein bands of PSD-95 and synaptophysin. (**B**) Protein quantification of PSD-95. (**C**) Protein quantification of synaptophysin. CA: 40 mg/kg; *n* = 3. Comparison with the WT group, ** *p* < 0.01; comparison with the 5xFAD group, ^#^ *p* < 0.05, ^##^
*p* < 0.01.

**Figure 6 ijms-25-03408-f006:**
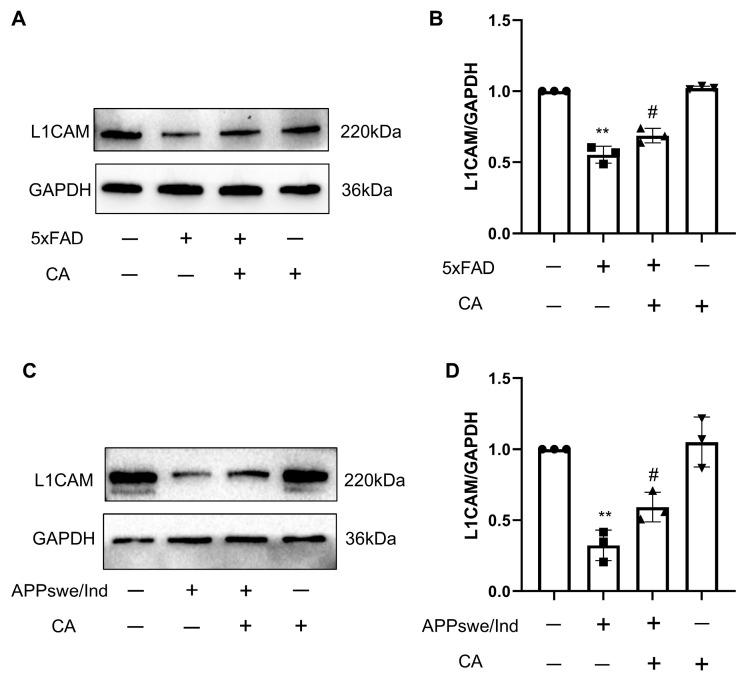
CA treatment increased L1 levels in APPswe/Ind transgenic SH-SY5Y cells and 5xFAD mice. (**A**) Protein bands of L1. (**B**) Protein quantification of L1. (**C**) Protein bands of L1. (**D**) Protein quantification of L1. *n* = 3. Comparison with the WT group, ** *p* < 0.01; comparison with 5xFAD mice, ^#^ *p* < 0.05.

**Figure 7 ijms-25-03408-f007:**
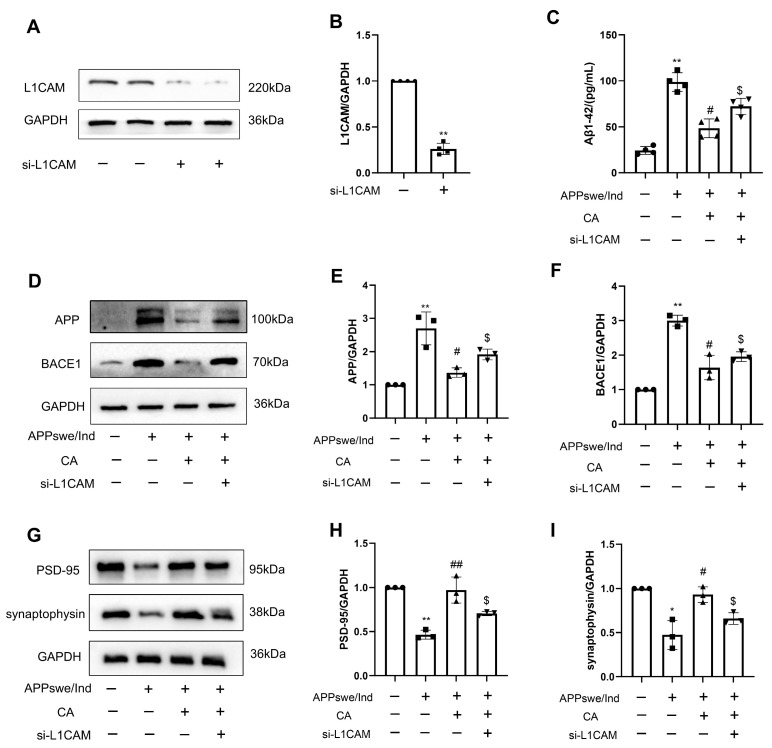
Influence of siL1-RNA on CA-treated APPswe/Ind transgenic SH-SY5Y cells. (**A**) Protein bands of L1. (**B**) Protein quantification of L1. (**C**) Levels of Aβ1–42. (**D**) Protein bands of APP and BACE1. (**E**) Protein quantification of APP. (**F**) Protein quantification of BACE1. (**G**) Protein bands of PSD-95 and synaptophysin. (**H**) Protein quantification of PSD-95. (**I**) Protein expression quantification analysis of synaptophysin. CA: 80 μM; *n* = 3–4. Comparison with the SH-SY5Y cells group, * *p* < 0.05, ** *p* < 0.01; comparison with the APPswe/Ind group, ^#^ *p* < 0.05, ^##^
*p* < 0.01; comparison with the APPswe/Ind + CA group, ^$^ *p* < 0.05.

**Table 1 ijms-25-03408-t001:** The primers of RT-PCR-detected genes.

Gene	Forward Primer	Reverse Primer
APP	5′-TGGAGGTACCCACTGATGGT-3′	5′-ACTGCATGTCTCTTTGGCGA-3′
BACE1	5′-CAGGCTTGTTCTTCACAGGG-3′	5′-ACCACAAAGCCTGGCAATCTC-3′

## Data Availability

The original contributions presented in this study are included in the article/Appendix A; further inquiries can be directed to the corresponding author.

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
