# Peer review of "Chicoric Acid Ameliorated Beta-Amyloid Pathology and Enhanced Expression of Synaptic-Function-Related Markers via L1CAM in Alzheimer’s Disease Models"

_ijms, 2024, doi:10.3390/ijms25063408_

Round 1
Reviewer 1 Report
Comments and Suggestions for Authors
The paper by Ruonan Wang et al. «Chicoric Acid Inhibits Beta-Amyloid Formation and Enhances Expression of Synaptic Function-Related Markers via L1CAM in Alzheimer's Disease Models» is an interesting study and the authors have collected a unique dataset. The authors conduct very relevant research.
Overall, the information presented represents valuable data on the analysis of the ability of chicoric acid (CA) to reduce APP and Aβ1-42 levels in SH-SY5Y cells and in a mouse AD model. The paper is generally well written. However, in my opinion, discussions of the findings are disputable and further clarification is required for confirmation of the scientific value of the work presented. Essentially, the authors suggest that the deposition of Aβ is the initial pathological event in the Alzheimer’s disease leading to the formation of neurofibrillary tangles (NFTs), cell death, and ultimately dementia. However, a huge body of evidence show that APP overexpression, leading to intensive Aβ production, is observed in multiple pathologies associated with the breakdown of energy metabolism in the brains both in humans and animals, and that it is necessary to fight not with amyloids, but with the reasons that “gave birth to them”. This view is supported by a large body of clinical trials showing negative effects, and even death in AD patients who received anti-amyloid preparations. Obviously, the authors need to explain this contradiction.
I feel it has potential for publication after more complete explanation of the point of view regarding this matter.
Author Response
Dear Reviewer,
We are also very grateful for your thoughtful and constructive comments that helped to improve the quality of our manuscript. The point-by-point responses for your comments are blow.
Comment:
Overall, the information presented represents valuable data on the analysis of the ability of chicoric acid (CA) to reduce APP and Aβ1-42 levels in SH-SY5Y cells and in a mouse AD model. The paper is generally well written. However, in my opinion, discussions of the findings are disputable and further clarification is required for confirmation of the scientific value of the work presented. Essentially, the authors suggest that the deposition of Aβ is the initial pathological event in the Alzheimer’s disease leading to the formation of neurofibrillary tangles (NFTs), cell death, and ultimately dementia. However, a huge body of evidence show that APP overexpression, leading to intensive Aβ production, is observed in multiple pathologies associated with the breakdown of energy metabolism in the brains both in humans and animals, and that it is necessary to fight not with amyloids, but with the reasons that “gave birth to them”. This view is supported by a large body of clinical trials showing negative effects, and even death in AD patients who received anti-amyloid preparations. Obviously, the authors need to explain this contradiction.
I feel it has potential for publication after more complete explanation of the point of view regarding this matter.
Authors’ Response:
We agree with the reviewer that the best way to prevent the formation of Aβ-42 and Aβ-40 and henceforth AD is a so-called healthy lifestyle. This prevention encompasses, for instance, a well-balanced physiological metabolism. However, despite a healthy lifestyle, dementia with aging cannot be prevented altogether. To find natural compounds that prevent formation of aggregates, we focused on the effect of CA influence on Aβ levels. CA is a natural compound that is readily found, for instance, in chicory salad, which is very popular in many countries. Therefore, we think that we do not have to justify more than we did already, namely that CA is an accepted natural compound. We would like to mention in this context that anti-amyloid treatments have so far been unsuccessful, including the use of monoclonal antibodies against Aβ peptides, although once FDA approved on questionable statistical evaluation. Therefore, pharmaceutical companies are now increasingly turning to compounds that prevent Aβ accumulation. We agree with the reviewer that accumulation of protein aggregates in many neurodegenerative diseases is the main reason for neurodegeneration. The prevention of this accumulation is therefore a central theme in such diseases. Please see page 8, lines 216-230.
Reviewer 2 Report
Comments and Suggestions for Authors
I have some comment on the manuscript entitled “Chicoric Acid Inhibits Beta-Amyloid Formation and Enhances 2 Expression of Synaptic Function-Related Markers via L1CAM 3 in Alzheimer's Disease Models” which are as follows.
Can you elaborate on the specific methods used to quantify Aβ1-42 expression in both the SH-SY5Y cells and the 5xFAD mice? Were there any challenges or limitations encountered during this process?
In the transcriptome sequencing analysis comparing changes in hippocampal gene expression between treated and untreated 5xFAD mice, how was the data normalized and analyzed to identify significant differences? Were any specific pathways or gene networks found to be particularly affected by chicoric acid treatment?
Could you explain the rationale behind choosing APPswe/Ind SH-SY5Y cells as a cellular model for Alzheimer's disease in this study? How well do these cells recapitulate the pathological features of AD, and what are their limitations in terms of representing the complexity of the disease?
Regarding the observed reduction in Aβ plaque levels following chicoric acid treatment, were there any investigations into the mechanism by which chicoric acid mediates this effect? For instance, does it directly interfere with Aβ aggregation or clearance pathways?
You mentioned that chicoric acid increased the expression of synaptic function-related markers such as L1CAM, PSD95, and synaptophysin. Can you discuss the significance of these markers in the context of Alzheimer's disease pathology and how their upregulation might contribute to improved synaptic function and cognitive outcomes?
In the siRNA-L1CAM experiment, how specific was the knockdown of L1CAM expression, and were there any off-target effects observed? Additionally, did the reduction in L1CAM expression completely reverse the effects of chicoric acid treatment on synaptic marker expression, or were there residual effects?
Considering the findings that chicoric acid affects synaptic plasticity-related genes in 5xFAD mice, do you have any insights into the potential clinical implications of these results for developing novel therapeutic interventions for Alzheimer's disease?
Were there any observations or analyses conducted to assess potential adverse effects or toxicity associated with chicoric acid treatment in either the cellular or animal models used in the study? If so, what were the findings, and how might they impact the translational potential of chicoric acid as a therapeutic agent for AD?
Include 2 relevant bibliographic studies PMID: 30605887 & PMID: 30503937 in your manuscript.
Given the multifactorial nature of Alzheimer's disease, do you believe that chicoric acid may have potential as a standalone therapeutic agent, or would it likely need to be used in combination with other drugs targeting different aspects of the disease pathology? What are the implications of this for future drug development strategies?
Could you provide more insight into the mechanism by which chicoric acid regulates APP-related secretases and synaptic function-associated proteins? Are there any specific pathways or signaling cascades that are known to be affected by chicoric acid treatment?
In terms of clinical translation, do you anticipate any challenges or considerations regarding the bioavailability and pharmacokinetics of chicoric acid as a potential therapeutic agent for Alzheimer's disease? Have there been any studies investigating the optimal dosing regimen or formulation to maximize its efficacy?
Given the observed reduction in Aβ levels and improvement in synaptic function-related markers, have you explored whether chicoric acid treatment leads to improvements in cognitive function or behavior in the 5xFAD mouse model? If so, what were the findings, and how do they contribute to the overall therapeutic potential of chicoric acid for Alzheimer's disease?
Are there any specific downstream targets or effectors of L1 expression that have been identified as mediators of chicoric acid's therapeutic effects in Alzheimer's disease? Understanding the molecular pathways involved could provide valuable insights for further mechanistic studies and therapeutic development.
Considering the complex etiology of Alzheimer's disease, do you envision chicoric acid being used as a monotherapy or in combination with other therapeutic agents targeting different aspects of the disease pathology? Have there been any investigations into potential synergistic effects when combining chicoric acid with existing Alzheimer's treatments?
Have there been any investigations into the long-term effects of chicoric acid treatment in the 5xFAD mouse model, particularly in terms of its impact on disease progression and neurodegeneration over time? Understanding the durability of chicoric acid's effects could be crucial for assessing its potential as a disease-modifying therapy for Alzheimer's disease.
Given that L1 expression appears to play a key role in mediating the therapeutic effects of chicoric acid, have you explored whether modulation of L1 expression alone is sufficient to replicate the beneficial effects observed with chicoric acid treatment? Understanding the specific contributions of L1 to the overall therapeutic mechanism could inform future drug development strategies.
Are there any potential off-target effects or unintended consequences associated with upregulation of L1 expression by chicoric acid? Have there been any investigations into the broader physiological or neurological effects of L1 modulation in the context of Alzheimer's disease?
How do the findings from your study align with previous research on chicoric acid and its potential therapeutic applications? Are there any discrepancies or areas of divergence that warrant further investigation or clarification?
Finally, do you plan to conduct further studies to validate and extend these findings, such as investigating the long-term effects of chicoric acid treatment on disease progression and cognitive function in animal models or clinical trials involving human subjects? If so, what specific research questions or hypotheses would you prioritize in future investigations?
Comments on the Quality of English Language
Complete editorial checking will be needed for the manuscript.
Reviewer 3 Report
Comments and Suggestions for Authors
The authors investigated the effect of chicoric acid in complementary preclinical models of Alzheimer's disease (AD) a mouse and a cell culture model. The main hypothesis is that the beneficial effect of CA treatment is mediated via the protein abbreviated as "L1" playing a central role in the amelioration of the AD phenotype. The presented results are convincing, the use of experimental techniques reflect a well-designed study plan. The results are clearly presented and easy to follow most of the time.
Minor concerns:
1. The use of CA in the study is a high single dose, no dose-response relationships are assessed. A detailed rationale of the secelcted dose and the limitation of no dose-response should be stated.
2. Figure 7 is very important to show the importance of L1. However, on this figure's graphs, the groups that were actually treated with CA are not indicated! That should be clarified
3. Please include a scale bar to the hippocampal section photomicrographs (Figure 2).
4. Some of the abbreviations are not detailed at first mention, but for instance in the methods, that is the last section in this journal. Please make sure that all abbreviations are detailed at first mention and then the abbreviation is used exclusively. Also, a list of abbreviations would be welcome.
Comments on the Quality of English Language
English style is overall pleasing, some typos, missing spaces should be edited.
Author Response
Dear Reviewer,
We are also very grateful for your thoughtful and constructive comments that helped to improve the quality of our manuscript. The point-by-point responses for your comments are blow.
Comment 1:
The use of CA in the study is a high single dose, no dose-response relationships are assessed. A detailed rationale of the secelcted dose and the limitation of no dose-response should be stated.
Authors’ Response:
CA is widely used as nutrition supplement in health food. In China, substances of the same origin as food and medicine are generally considered non-toxic. In the animal experiment, we chose 40 mg/kg according to the references (Chicoric Acid Prevents Neuroinflammation and Neurodegeneration in a Mouse Parkinson's Disease Model: Immune Response and Transcriptome Profile of the Spleen and Colon; DOI:10.3390/ijms23042031). In this experiment, we first verified the pharmacodynamics of CA in 5xFAD mice, so only one concentration was used. However, we appreciate your comments. In the future study, we will set different dosage groups to verify the dose-dependence effects of the drug. We have added references to the basis for the selection of 40 mg/kg CA. Please see page 10, lines 339-340.
Comment 2:
Figure 7 is very important to show the importance of L1. However, on this figure's graphs, the groups that were actually treated with CA are not indicated! That should be clarified.
Authors’ Response:
We have added the description of CA treatment in Figure 7.
Comment 3:
Please include a scale bar to the hippocampal section photomicrographs (Figure 2).
Authors’ Response:
We have inserted a scale bar in Figure 2.
Comment 4:
Some of the abbreviations are not detailed at first mention, but for instance in the methods, that is the last section in this journal. Please make sure that all abbreviations are detailed at first mention and then the abbreviation is used exclusively. Also, a list of abbreviations would be welcome.
Authors’ Response:
With revisions, we have made sure that all abbreviations are detailed at the first mention and then used abbreviations exclusively. We have added the Abbreviations. Please see page 12, lines 452-470.
Abbreviations
AD Alzheimer's disease
Aβ amyloid-beta
APP amyloid precursor protein
Swe/Ind Swedish/Indiana
BACE1 β-secretase
CA chicoric acid
MPTP N-methyl-4-phenyl-1,2,3,6-tetrahydropyridine
PSD-95 post-synaptic density protein 95
L1CAM L1 cell adhesion molecule
WT wild-type
PBS phosphate buffered saline
RIPA Radio Immunoprecipitation Assay Lysis Buffer
SDS-PAGE sodium dodecyl sulfate - polyacrylamide gel electrophoresis
PVDF polyvinylidene fluoride
HRP horseradish peroxidase
ECL enhanced chemiluminescence
qRT-PCR quantitative real-time polymerase chain reaction
ELISA enzyme-linked immunosorbent assay
Round 2
Reviewer 1 Report
Comments and Suggestions for Authors
According to the literary data it must be acknowledged that data on toxicity of amyloid peptides are obtained exclusively from in vitro experiments, while participation of amyloids in neuronal death has not yet been demonstrated in vivo. This view is supported by a large body of clinical trials showing negative effects, and even death in sporadic AD patients who received anti-amyloid preparations, despite their therapeutic effect having been confirmed in transgenic mice.
And furthermore, it is confirmed by a huge body of evidence showing that APP overexpression, leading to intensive Аβ production, is observed in multiple pathologies associated with the breakdown of energy metabolism in the brains both in humans and animals. Thus, enhanced amyloid formation and accumulation occurs in the brains of people of different ages during: hypoxia, human and animal head trauma, spinal cord injury, neuroinflammation, middle cerebral artery occlusion, general anesthesia, influence of bacterial agents and other examples.
In this regard, in order to overcome the contradictions that arise when reading the discussion of the results, the authors need to explain this contradiction with use of modern literary data indicating that “Chicoric acid improves neuron survival by promoting mitochondrial function and energy metabolism” (PMID PubMed: 31501849, 35216146) supporting brain cell viability and its myriad functions.
These recommendations have not been taken into account by the authors. Without these conditions, the article cannot be published.
Author Response
Thank you for your comments. We are sorry not understanding your comments well in the first revision. Based on your kind advice, we rewrote the Discussion section.
Reviewer 2 Report
Comments and Suggestions for Authors
Authors are suggested to go through each and every suggestions suggested in my previous comments and revise the manuscript thoroughly.
Provide exact p value for all your histograms.
Provide complete western blot, not the cut one.
Scale bar should be included in all the groups of figure no-2.
The western blot of figure no.4 shows the dimer formation for PSD-95, repeat the WB.
Authors are suggested to include the therapeutic potential of ursolic acid and chlorogenic acid in MPTP and Rotenone intoxicated Parkinsonian mouse model in the manuscript. Because ursolic acid shows similiar response to Chicoric Acid.
Discussion section lacks coherence with the already published articles, revise this section.
Complete editorial checking is still needed for your manuscript.
Comments on the Quality of English Language
Complete editorial checking is still needed for your manuscript.
Round 3
Reviewer 1 Report
Comments and Suggestions for Authors
I am proposing you this article for publication.
Reviewer 2 Report
Comments and Suggestions for Authors
The manuscript has been revised as per my suggestions.